# How Much Do Front-Of-Pack Labels Correlate with Food Environmental Impacts?

**DOI:** 10.3390/nu15051176

**Published:** 2023-02-26

**Authors:** Luca Muzzioli, Lorenzo Maria Donini, Matteo Mazziotta, Marco Iosa, Francesco Frigerio, Eleonora Poggiogalle, Andrea Lenzi, Alessandro Pinto

**Affiliations:** 1Department of Experimental Medicine, Sapienza University, 00185 Rome, Italy; 2Italian National Institute of Statistics, 00184 Rome, Italy; 3Department of Psychology, Sapienza University, 00185 Rome, Italy; 4Santa Lucia Foundation, IRCSS, 00179 Rome, Italy

**Keywords:** front-of-pack, nutrient profiling, FOPLs, environmental sustainability, healthy and sustainable diets, global warming, climate change, composite indicator

## Abstract

Nutrient profiling and front-of-pack labeling (FOPL) have been developed to categorize food products as more or less healthy based on their nutrient content and to easily communicate this information to consumers. The goal is to change individual food choices toward a healthier diet. Since global climate change has recently become an urgent matter, this paper also aims to investigate the correlations between different food health scales, including some FOPLs currently adopted by one or more countries, and several sustainability indicators. For this purpose, a food sustainability composite index has been developed to summarize environmental indicators and compare food scales. Results indicate, as expected, that well-known healthy and sustainable diets are strongly correlated with both environmental indicators and the composite index, while FOPLs based on portions or on 100 g show moderate and weak correlation values, respectively. Within-category analysis has not found any associations that explain these results. Hence, 100 g standard, on which FOPLs are usually developed, seems not to be the ideal basis for developing a label that aspires to communicate healthiness and sustainability in a unique format, as required by the need for simple messaging. On the contrary, FOPLs based on portions appear to be more likely to achieve this goal.

## 1. Introduction

Despite the majority of developed countries acknowledging a high priority for health and health-related policies, in 2019, non-communicable diseases (NCDs) (e.g., cardiovascular disease, cancer, obesity, and type 2 diabetes) were responsible for 42 million deaths per year worldwide [1]. Most NCD-related deaths could be avoided by adopting a healthy lifestyle, including healthful eating patterns (e.g., the Mediterranean diet) [2]. Nutrition labelling has been one of the tools developed over the last three decades to promote healthy food choices. Nutrition labelling is an endeavor to give consumers, directly at the point of purchase (PoP), detailed information about the nutritional content of single food products, in order to enable them to make food choices that are nutritionally appropriate. Potentially, it could be a powerful instrument that supports the aim of healthy eating while retaining consumer freedom of choice and reducing the time need for consumers to search for the required information [3].

In recent years, considering that food companies previously confined nutritional information to the back of the packaging (BOPLs), a new form of food labelling called front-of-pack labels (FOPLs) has been specifically developed as a healthy tool for nutritional education and/or containment of NCDs. Unlike BOPLs, which are commonly considered difficult to understand and time-consuming [4], especially at the PoP, FOPLs were created as a simple message aimed to facilitate individual food choices for a healthful dietary pattern.

Several different types of FOPLs have been proposed, and some countries have already adopted these. A first classification divides them into directive and non-directive labels. Directive labels use an algorithm that warns and/or ranks foods, with the goal of modifying individual eating behaviors by associating each food with different levels of healthfulness [5]. The most commonly used models of directive labels are ranking labels, traffic light labels, warning labels, and endorsement logos. By contrast, non-directive FOPLs (such as reference intakes and related labels), are informative labels that rely solely on nutrition facts. They show the amount of a small number of food components (typically fat, saturated fats, sugars, salt, and energy) in the food and relative (as percentages) amounts within a standard daily dietary pattern and, in some cases, also display the amounts that refer to a recommended serving size. Therefore, they do not associate any judgment with foods, and leave the consumer free to interpret the information provided. [6] Given these characteristics and their increasing acceptability, the European community is currently debating the adoption of a single FOPL among EU members in order to harmonize the market’s current multiplicity. However, the findings from various studies indicate that attempts to change individual behaviors in order to improve eating patterns achieve only limited success [7]. FOPLs, despite being better accepted and understood by consumers than BOPLs, have shown a similar limited capability of modifying individual food choices so far [8].

It is well known that the consumption of food contributes to a significant proportion of a person’s overall greenhouse gas (GHG) impact, such that food systems account for 34% of global anthropogenic GHG emissions, to which agriculture and land use/land-use change activities are the largest contributors [9]. For these reasons, individual dietary choices have become crucial in lowering consumer-driven emissions, and many attempts are being made towards a new definition of dietary eating patterns that incorporates environmental sustainability [10,11]. Some dietary patterns have already demonstrated a strong link between health and sustainability; for instance, the Mediterranean diet (MD) is the most well-known and well-studied example of a healthy dietary pattern with a low impact on the main environmental impact indicators [12]. Lastly, considering that in recent years consumers have displayed a moderately high level of concern for sustainability issues but a low level of engagement in the context of concrete food product choices driven by environment-specific labels [13], every tool targeted at the achievement of a healthier individual diet should also focus on the attainment of a more sustainable diet.

The aim of this study is to understand whether, and to what extent, current FOPLs integrate healthiness and sustainability. In order to do so, this paper investigates the correlations between different food health scales, including some FOPLs currently adopted by one or more countries, and several sustainability indicators, such as carbon footprint, land use, and water footprint.

## 2. Materials and Methods

### 2.1. Study Design

The study design utilized a stepwise approach. Firstly, a literature search has been conducted to collect food items from studies that assessed food life cycle assessments (LCAs) for several environmental indicators. From the sum of all identified foods, a final food list was extrapolated by removing duplicates and nonmatching foods with a food composition database. Then, eligible FOPLs and food scales were identified and, according to their underpinning algorithms, ranks were calculated for all listed foods to convert them into food scales. Lastly, a composite index was developed to summarize environmental indicators and assess correlations with food scales.

### 2.2. Food Product Selection

The first stage of the research was to search the PubMed database for articles focused on LCAs of food commodities published in the last 5 years. LCA is the process of evaluating the environmental impact of a commodity over its lifetime [14]. The following searching string was used in order to select pertinent systematic reviews and meta-analyses: food AND—followed by other sets of words—global warming potential, greenhouse gas emissions, environmental impact, life cycle, LCA. Article selection was carried out in June 2022 and three papers were identified [14,15,16].

The following passage consisted of an overall grouping of all food items listed in the aforementioned manuscripts, which resulted in a total of 500 food items selected. Then, the list was examined to eliminate all beverages and non-edible commodities. Lastly, the detection and the elimination of duplicate items between the different papers were carried out, resulting in a final list of 182 food commodities. The food selection flow chart is shown in Figure 1.

The final food list collected the impact values of eight indicators of environmental sustainability: three concerning carbon emissions (measured as kg CO_2_ eq/kg), namely, global warming potential (GWP) [14], greenhouse gas emissions (GHG) [15], and carbon footprint [16]; two indicators for water consumption (measured in LH_2_O/kg), called water withdrawal [15] and water footprint [16], and one each on land use (measured as m^2^/kg), acidification (measured as g SO_2_ eq/kg), and eutrophication (measured as g PO43- eq/kg), all assessed by the method described by Poore et al. [15]. For LCA, all the units of measurement are reported as functional units. These functional units are typically based on the primary function that a product or service provides; for instance, in the study by Poore et al. [15], units are expressed as retail weight functional units (FU) and include losses between distribution and retail but not consumer losses. Functional units are converted into kg for solid commodities and into mL for liquid ones, as indicated in the present research.

### 2.3. Nutrient Profiling and FOPL Selection

After the finalization of the food list, a search for nutrient profiling was carried out among those that are currently adopted by countries. Given the nature of the final list, where most of the foods were unprocessed or minimally processed, all nutrient profiling and/or FOPLs applicable uniquely to processed foods (e.g., the Chilean warning labels) have not been taken into consideration, nor have the endorsement logos (e.g., Smart Choices, Nordic Keyhole), which classify foods in a binary “yes” or “no” modality, impeding correlation analysis. On the other hand, FOLPs that divide foods into at least four groups, ranging from less to more healthy (or, similarly, from those that should be consumed in the least amount to those that should be consumed in greater amounts) have been included in the study.

The Nutri-Score (NS), the Health Star Ratings (HSR), the Health Canada Surveillance Tool (HCST), the NutrInform Battery (NIB), and the Israeli food labels (IFLs) were selected for this study. According to a recent article by Muzzioli et al. (2022), NS and HSR are classified as directive labels, IFL as semi-directive, and NIB as non-directive [6]. HCST is currently a nutrient profiling tool with no plans to be applied to food packaging. From the point of view of the algorithm, NS, HSR, and IFLs have been developed on a 100 g standard basis, while HCST and NIB are based on reference portions in accordance with their respective national dietary guidelines. Then, food ranks were calculated for each FOPL, as explained in Section 2.5.

### 2.4. Healthy and Sustainable Diet Selection

In order to validate the results of the correlations between FOPLs and the environmental indicators, it was important to compare them to food scales that already include both healthiness and sustainability. Thus, as a positive control of the research, validated healthy and sustainable diets, such as the Italian National Dietary Guidelines recommended diet (ITA-GL) [17], the EAT-Lancet Commission diet (EAT Lancet) [11], and the Mediterranean diet (MD) [12], have been selected and transformed into food scales (as described in Section 2.6).

### 2.5. Food Rank Calculation

Once the selection of FOPLS and heathy diets was completed, the next stage was aimed at calculating the ranks of listed foods to obtain a food scale for each FOPL. Ranks were calculated according to the last official release of each FOPL, the last published version of the user manual, the current guidelines, or the articles in which they are explained. The composition of foods’ nutrients was assessed using the open access database “Food Composition Database for Epidemiological Studies in Italy” (BDA-IEO). Nutrients taken into consideration in the present study were sugars, saturated fatty acids (SFA), proteins, sodium, fibers, total fats (TF), and energy, calculated on 100 g of the edible part of the food product.

Nutriscore: Nutri-score scores were computed following the Nutriscore Frequently Asked Questions document [18] and the Nutriscore calculator file [19]. The resulting ranks, ranging from A to E, were assigned a score from 1 to 5. It should be noted that, at the time of writing, the developers had just released an updated version of the Nutriscore, which was not considered in this paper.

Health Star Ratings: scores were computed according to the HSR user manual [20] and the HSR calculator file [21]. The resulting ranks from 1 to 5 stars were transformed into a 1-to-10 scale, considering the half-star sensitivity of the HSR scale.

Health Canada Surveillance Tool: foods were divided into 4 ranks (tiers), according to the official Health Canada document [22]. Foods were classified into four categories, ranging from those recommended in higher amounts (Tier 1) to those recommended in lower amounts (Tier 4). A 5th group was added to gather some foods not included in the 4 tiers but that should be eaten in smaller amounts: “high fat and/or high sugar foods (e.g., foods high in sugar and/or fat that could not be assigned into one of the four major food groups, such as candies, chocolate, syrups, sauces, etc. as well as high fat/sugar foods that are usually eaten in small quantities, i.e., not large enough to contribute to a Food Guide Serving)”. Oils and fats were not included in any group, as mentioned in the document.

NutrInform Battery: the food ranks were calculated using the official website’s computation: firstly, the amount per 100 g of energy and four nutrients (total fats, saturated fats, sugars, and salt) of each food item was multiplied by the recommended portions displayed by the official smartphone app. The relative amounts were then expressed as a percentage of the balanced 2000 Kcal reference diet. Finally, the final scores were calculated by adding the five relative percentages, yielding a scale from 0 to 500.

Israeli food labels: scores were calculated on a 5-range scale from −1 to 3, considering the application of a green label as a −1 score, no label as a 0 score, and the possible combinations of the 3 red labels as a 1, 2, or 3 score, respectively. Labels were attached according to Gillon-Keren et al. [23] and the thresholds used for the label assignment were based on the second stage (January 2021).

### 2.6. Healthy and Sustainable Diet Food Rank Calculation

In contrast to FOPLs, healthy diets were transformed into food scales following the principle that the more the recommended amount of a food, the more it can be considered healthy. This is in accordance with what was applied by Scarborough et al. [24] in the development of the FSA-NPS, which is the basis of the Nutriscore and the HRS labels “[…] potential respondents were told that, for the purposes of the survey, a ‘healthier’ food was a food which should be eaten frequently and/or in large amounts by a person aiming to meet public health nutrition recommendations, and conversely that a ‘less healthy’ food was a food which should be eaten infrequently and/or in small amounts […]”.

CREA Italian Dietary Guidelines: weekly amounts (in g) of foods cited in the CREA guidelines were obtained by multiplying the recommended serving size by the recommended frequency [17] (pp. 131–141). Weekly relative food amounts were used as ranks on a 25–3500 g scale (from the minimum to the maximum value).

EAT-Lancet Commission Diet in the Anthropocene: daily amounts (in g) were extracted from Willet et al. [11] (p. 451) and used as ranks on a 7–300 g range.

Mediterranean diet: ranks were assigned from 1 to 6 according to the different assignments of food groups on the pyramid levels, starting from the lower to the upper level, as depicted in the updated Mediterranean diet pyramid [12].

### 2.7. The Development of a Composite Index

The last stage consisted of the development of a composite index to reduce the multidimensional nature of sustainability by summarizing the eight environmental indicators into a single vector. The reduction of dimensions through the development of a single index allows the quantification, and therefore rank, of the environmental impact of diets and FOPLs. The index was designed according to the Mazziotta Pareto Index protocol [25] and called the Food Sustainability Mazziotta Pareto Index (FS-MPI). Data first underwent a normalization stage through a modified z-score. The normalized dataset was then aggregated by the sum of the arithmetic mean with a penalty function derived by the product of the standard deviation and the coefficient of variation.

### 2.8. Data Analysis

Data entry and analysis were carried out with the statistical software Jamovi (Ver. 2.3.18). A descriptive statistics analysis was carried out to study the behavior of all the variables, and then the Spearman correlation was applied to find possible correlations between the 16 variables in the object. In addition, a hierarchical cluster analysis (parameter settings: distance measure = Euclidean; clustering method = warp D2) was performed on the correlation matrix to individuate patterns among food scales that could explain the observed relationships with environmental indicators. Finally, the FS-MPI was applied to the 8 environmental indicators to rank the overall environmental impact of diets and FOPLs using the composite index and assess which scale had the strongest correlation.

## 3. Results

### 3.1. Foods Included in the Analysis

As depicted in the flow chart (Figure 1), 182 different types of food were investigated. Table 1 divides these foods in 15 main categories (the majority of these foods are raw or unprocessed). A detailed list of all commodities can be found in Appendix A. As shown in Table 1, the number of foods included among categories varies between 33 and 1, with fruits having the highest number and eggs having the fewest. Despite this wide range, sources of protein, carbohydrates, fat, and dietary fiber are well balanced, with 76, 72, 51, and 56 food commodities, respectively. The use of the BDA-IEO database impeded the inclusion of some foods culturally and geographically distant from the Mediterranean and, more generally, European dietary patterns.

### 3.2. Food Scale Correlation Analysis

For each of the eight scales of food healthiness (computed for the above 182 foods), the correlation with each one of the eight indicators of environmental sustainability was computed according to the studies of Clune et al. (global warming potential) [14], Poore et al. (greenhouse gas emissions, land use, acidification, eutrophication, and freshwater withdrawal) [15], and Petersson et al. (carbon footprint and water footprint) [16]. The obtained correlation matrix is reported in Table 2.

Healthier foods are associated with higher values of ITA-GL, Eat Lancet diet, and HSR, and with lower values for MD, NS, IFLs, NIB, and HCST. This explains the reason of some inverse (negative) correlations with the environmental indicators, for which higher values correspond to higher environmental impacts. The last column and the last raw data report the means (and standard deviations) of the absolute values of these correlations.

The eight food scales showed different correlations, ranging, as absolute values, from a maximum of 0.824 to a minimum of 0.030. Five scales showed an average correlation above 0.5. The majority of the correlations were found to be statistically significant.

### 3.3. Food Scales Correlation Patterns

Figure 2 provides different visual inspections of the values reported in the correlation matrix in Table 2. In Figure 2a, a radar chart was carried out from the eight emission indicators correlated to the eight food scales mapped. External lines correspond to higher correlations and as shown in the radar chart, three lines highlight strong correlations (MD, ITA GL, and EAT Lancet), two lines have moderate correlations (NIB and HCST), two lines with low correlations (NS and IFLs), and the most internal line has weak correlation (HSR). A similar pattern emerges from the heatmap of the cluster analysis, in Figure 2b, which was performed to understand which food scale might have similar behavior with respect to the environmental indicators. Moving from left to right (brighter colors indicate lower absolute correlation values), there is HSR, followed by NS and Israeli WLs, then in the middle, HCST and NIB, and the healthy and sustainable diet on the right side of the map.

Finally, the same division between the eight food scales emerges also into the dendrogram (Figure 2c), which shows the relationships among the eight food scales, where four subgroups are depicted: MD, ITA GL, and ELD (in green); NIB and HCST (in blue); NS and IFLs (in yellow); and the HSR (in grey).

### 3.4. Food Scale Group Correlations Analysis

Based on the results of the cluster analysis and the intrinsic characteristics of each of the selected food scales, another evaluation was then carried out by dividing the eight food scales into three groups and calculating the average of the Spearman’s correlations reported in Table 2 for each group. The three groups were: healthy diets (HDs, comprising MD, ITA GL, and EAT Lancet), food scales based on portions (FOPL/portions, NIB and HCST), and FOPLs based on 100 g reference (FOPLs/100 g, NS, IFLs, and HSR). In Figure 3, it can be observed that HDs correlated better, on average, for all indicators, reaching a maximum in GWP (HDs = 0.762). Between the two FOPL groups, FOPLs/portions showed higher correlations than FOPLs/100 g for all the indicators. Finally, FOPLs/100 g not only showed lower correlation values, but also higher variability (as highlighted by the wide standard deviations).

### 3.5. The Food Sustainability Mazziotta–Pareto Index Analysis

The last step of the analysis was carried out using the Food Sustainability Mazziotta–Pareto Index (FS-MPI) to summarize the eight environmental indicators into a single index. Even though the Mazziotta–Pareto Index is usually applied in the economics and human development fields of research [25,26], its solid background in the scientific literature makes it preferable to the new environmental index recently developed by Clark et al. [27]. However, if compared in a ranking order, the positions of the scales are the same with the two methods.

Spearman correlations between FS-MPI and the eight food scales are shown in Figure 4. All of them are statistically significant, but the Mediterranean diet showed the highest score (R = 0.738), then the EAT-Lancet and the Italian Dietary Guidelines with similar results (0.705 and 0.676, respectively), followed by NIB (0.497) and HCST (0.431), then the Israeli WLs and NS with very narrow differences (0.379 and 0.353, respectively), and, lastly, HSR (0.185) (Figure 4).

## 4. Discussion

Despite the great attention that FOPLs have gained in the last few years and their current and future use, only one previous research study has investigated their relationship with the environmental impact of various foods [27]. With the current crisis of climate change, it is important to thoroughly investigate this possible relationship.

Our research focused on identifying possible correlations between four FOPLs (Nutriscore, HSR, Israeli FLs, and NutrInform Battery), one nutrient profiling tool (HCST), and eight food environmental indicators (GWP, land use, GHG, acidification, eutrophication, freshwater withdrawal, CO_2_ footprint, H_2_O footprint) aggregated from three different studies [15,16,17]. Moreover, three well-known healthy and sustainable diets (the Mediterranean diet, the diet recommended by the Italian Dietary Guidelines, and the EAT-Lancet diet in the Anthropocene) were also investigated.

The findings indicate a high correlation between the eight environmental indicators and the three healthy and sustainable diets (R ranged between 0.4 and 0.8), pointing to an overall similarity in results between the diets. These positive results were expected as the dietary patterns that were studied were developed either for their healthiness or for their low impact on the environment [11,28,29,30]. For instance, the Mediterranean diet was first recognized to improve life expectancy and to reduce disability-adjusted life years (DALYs) and then discovered to have a limited environmental impact [28,29]. The EAT-Lancet diet, on the other hand, was purposefully developed by a panel of technical experts to combine healthiness and a very low environmental impact as a scientific proposal to reduce food-driven emissions, responding to the growing climate change emergency [11]. Lastly, even the 2018 Italian Dietary Guidelines, which were developed to improve sustainability behavior of consumers [30], showed overlapping results in amplitude and variability with the other diets, especially with the EAT-Lancet diet. Despite the differences in macronutrient percentages between the EAT-Lancet diet and the Italian Dietary Guidelines, the similarities observed in Table 2 may come from the fact that they both recommend low amounts of animal products and that their food scales are both computed by the product of food portions and food frequencies. Thus, the results of the dietary patterns confirmed the validity of the computation method and the results obtained by the other food scales, whose environmental impact has not been investigated until now.

On the other hand, FOPLs showed low-to-modest correlations (0.2–0.5) with high data variability. Hence, not only do these results suggest a lower overall correlation strength of the nutrient profiling tools against dietary patterns, but they also indicate the numerous differences inherent in the nutrient profiling algorithms themselves. For example, IFLs and NS were found to score similarly in averages, either in the correlation matrix (0.382 ± 0.093 and 0.373 ± 0.106, respectively) or in the FS-MPI analysis (0.379 and 0.353, respectively). This could be because they were developed on populations that share the Mediterranean Sea, namely the French and the Israelis. Conversely, the HSR showed very low correlations with the eight environmental indicators investigated. This result cannot be entirely ascribed to low sustainability per se; a possible explanation might come from the long distance, either geographically or dietetically, between, on the one hand, Australia and New Zealand, where the label was developed and adopted, and, on the other hand, the European and Mediterranean areas, whose foods the BDA-IEO is mainly focused on.

Lastly, the result of the NIB and the HSTC opens numerous considerations. These classifications displayed the highest correlation with environmental indicators among food scales, as well as the best overall correlation with the freshwater withdrawal indicator (0.553 and 0.545, respectively). The dynamic is confirmed by cluster analysis and the FS- MPI, where NIB and HCST ranked with values between dietary patterns and the other FOPLs. The NutrInform Battery was developed with the purpose of helping consumers make more informed choices by showing the suggested portion, the amounts per portion, and the relative percentages of selected nutrients compared to a reference diet. Moreover, even if Canada is distant from Europe, as mentioned for Australia and New Zealand, the main similarity with NIB and, on the other hand, the main difference with the other FOPLs, comes from the fact that HCST considers reference amounts instead of the 100 g standard for dividing food into the four tiers. That reason could explain why NIB and HCST produce results closer to dietary patterns, in which foods are recommended based on frequency and serving size.

The analysis of food categories correlations (Appendix A), even if showing some significant correlation (bakery products, meats, milk and dairy, pulses), did not reveal any solid evidence that can help explain the correlation differences observed between food scales (healthy diets vs. FOPLs). This fact strengthens the hypothesis, pointed out for NIB and HCST, that the standard adopted (portions or 100 g) in developing a food healthiness scale may be one of the principal causes for the very low-to-moderate correlations observed in FOPLs with environmental indicators.

### Strengths and Limitations

The strength of this paper is to be, to the best of our knowledge, the first to compare several FOPL algorithms by finding their correlation with all the food environmental emission values published to date, aggregated from various studies. Furthermore, what this study adds to the literature is a quantitative way to compare FOPLs to each other, with diets, and with sustainability indicators.

This study has potential limitations. Firstly, the three papers on food LCAs [15,16,17] often evaluated subgroups of foods (e.g., root vegetables) instead of single food items, making it difficult to easily compare different studies. Moreover, due to intrinsic limitations of the BDA-IEO database, the number of food items included in our final analysis was 182. A higher number may have generated more reliable findings.

## 5. Conclusions

Environmental sustainability has become an increasingly urgent topic, and current global efforts are insufficient to limit warming to 1.5 °C [31]. This paper indicates that, in the field of food labels, efforts are directed towards a partial objective that views healthiness as the dominant characteristic of our diet. However, this approach is too narrow. Based on the present findings, a single-perspective approach is no longer consistent with the present situation, and FOPLs and nutrient profiling should now embed environmental sustainability as a key component in their development. Our research found some evidence for the importance of including frequency and/or serving size in nutrient profiling algorithms; they demonstrated a generally better correlation with major environmental indicators, such as complete dietary patterns, the NutrInform Battery, and the Health Canada Surveillance Tool. Algorithms developed on a 100 g basis as reference standards, on the other hand, lacked strength in correlations across all environmental indicators.

Thus, the debate proposed by Scarborough et al. [32] on the design of front-of-pack labels needs to be reopened, since we, as scientists, are now required to modify our tactics [33] and adopt a multidisciplinary approach. In particular, a major goal of diet recommendations is to encourage the population to adopt food choices that have a higher environmental impact.

## Figures and Tables

**Figure 1 nutrients-15-01176-f001:**
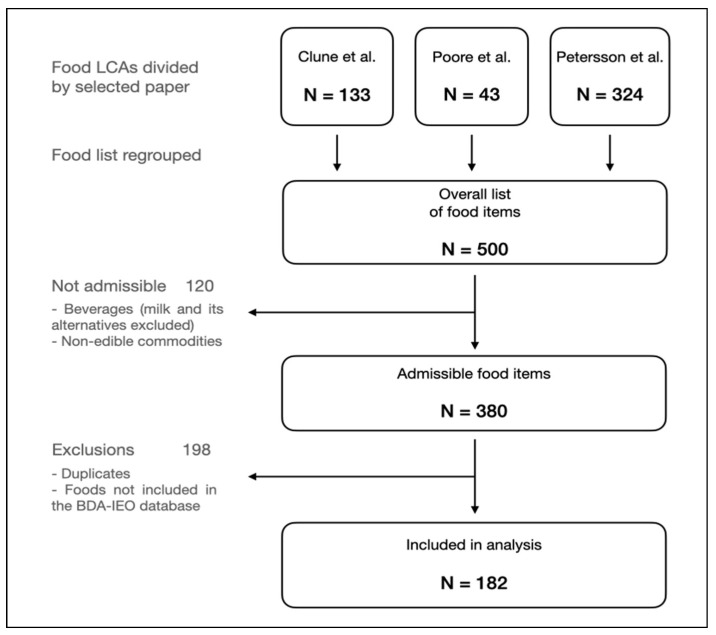
Flow chart of the food product selection. Abbreviations: LCAs = life cycle assessments [14,15,16].

**Figure 2 nutrients-15-01176-f002:**
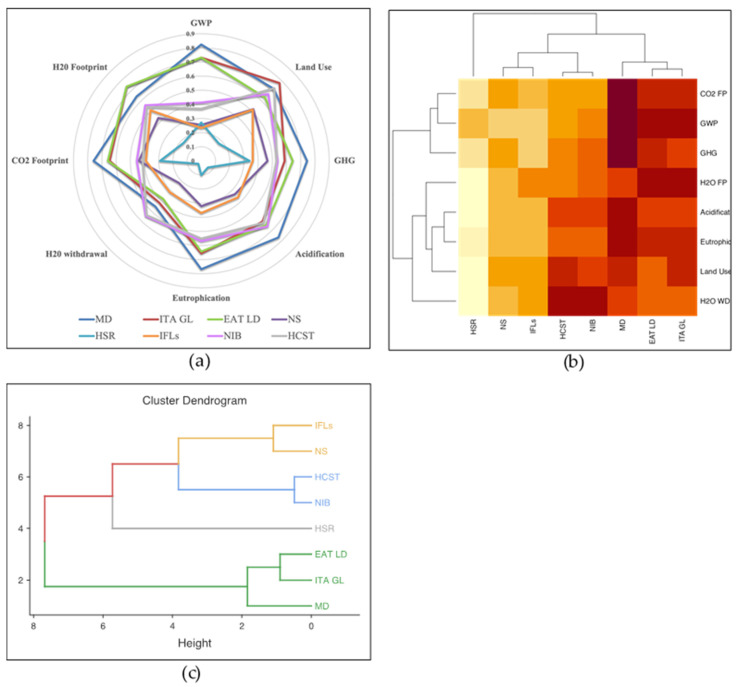
(**a**) Radar chart of Spearman’s correlations between FOPLs, healthy and sustainable diets, and indicators of environmental sustainability; (**b**) heatmap of cluster analysis between FOPLs, healthy and sustainable diets, and indicators of environmental sustainability; (**c**) dendrogram on FOPLs and healthy and sustainable diets. Abbreviations: MD = Mediterranean Diet; ITA GL = Italian Dietary Guidelines; EAT LD = EAT Lancet Diet; NS = Nutriscore; HSR = Health Star Ratings; IFLs = Israeli Food labels; NIB= NutrInform Battery; HCST = Health Canada Surveillance Tool; CO_2_ FP = CO_2_ Footprint; GWP = Global Warming Potential; GHG = Greenhouse Gas; H_2_O FP = H_2_O Footprint; H_2_O WD = H_2_O withdrawal.

**Figure 3 nutrients-15-01176-f003:**
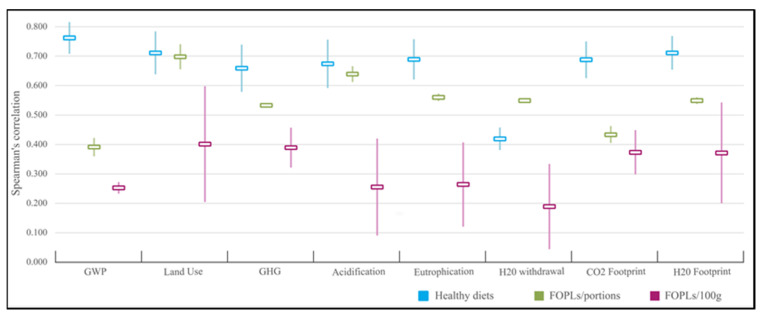
Average of Spearman’s correlations for healthy diets (in blue), FOPLs based on portions (in green), and FOPLs based on 100 g (in purple) (standard deviations are shown in lighter colors). Abbreviations: GWP = global warming potential; GHG = greenhouse gas emission.

**Figure 4 nutrients-15-01176-f004:**
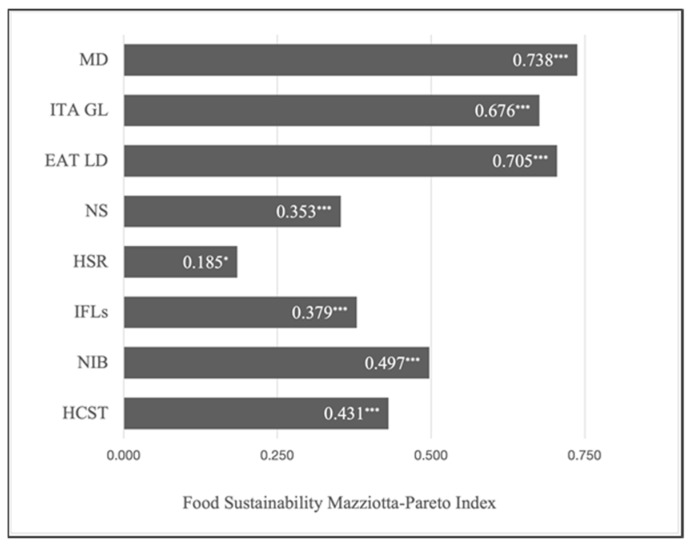
FOPLs and healthy and sustainable diets comparison using the Food Sustainability Mazziotta–Pareto Index as a composite index for the 8 sustainability indicators. Abbreviations: MD = Mediterranean Diet; ITA GL = Italian Dietary Guidelines; EAT LD = EAT Lancet Diet; NS = Nutriscore; HSR = Health Star Ratings; IFLs = Israeli Food labels; NIB = NutrInform Battery; HCST = Health Canada Surveillance Tool. ***: *p* < 0.001; *: *p* ≤ 0.05.

**Table 1 nutrients-15-01176-t001:** Food commodity division into food categories.

Category	N° of Items	Category	N° of Items	Category	N° of Items
Fruits	33	Milk Alternatives	4	Tubers	4
Vegetables	23	Pulses	11	Oils and Fats	10
Grains	10	Nuts and seeds	10	Sweets and Desserts	6
Meats	8	Processed Meat	3	Bakery products	9
Milk and Dairy	20	Eggs	1	Fish and Seafood	30

**Table 2 nutrients-15-01176-t002:** Correlation matrix between food scales and environmental sustainability indicators.

	GWP	Land Use	GHG	Acidification	Eutrophication	H_2_O Withdrawal	CO_2_ Footprint	H_2_O Footprint	μ ± σ
MD	0.824 ***	0.721 ***	0.745 ***	0.767 ***	0.768	0.458 *	0.755 ***	0.644 ***	0.710 ± 0.114
ITA GL	−0.731 ***	−0.779 ***	−0.587 ***	−0.610 ***	−0.656 ***	−0.419 *	−0.646 ***	−0.744 ***	0.647 ± 0.114
EAT Lancet	−0.731 ***	−0.634 ***	−0.644 ***	−0.645 ***	−0.643 ***	−0.381 *	−0.658 ***	−0.744 ***	0.635 ± 0.111
NS	0.254 **	0.517 **	0.466 **	0.334	0.321	0.222	0.439 ***	0.429 ***	0.373 ± 0.106
HSR	−0.271 **	−0.174	−0.339	−0.066	−0.101	−0.030	−0.291 ***	−0.179 *	0.181 ± 0.112
IFLs	0.232 *	0.513 **	0.362 *	0.366 *	0.369 *	0.314	0.390 ***	0.506 ***	0.382 ± 0.093
NIB	0.413 ***	0.668 ***	0.537 ***	0.658 ***	0.569 ***	0.553 **	0.454 ***	0.557 ***	0.551 ± 0.088
HCST	0.369 ***	0.728 ***	0.530 **	0.620 ***	0.551 **	0.545 **	0.413 ***	0.541 ***	0.537 ± 0.112
μ ± σ †	0.478 ± 0.244	0.592 ± 0.194	0.526 ± 0.137	0.508 ± 0.232	0.497 ± 0.218	0.365 ± 0.175	0.506 ± 0.160	0.543 ± 0.184	

Abbreviations: MD = Mediterranean diet; ITA GL= Italian Dietary Guidelines; NS = Nutriscore; HSR: Health Star Ratings; IFLs = Israeli food labels; NIB = NutrInform Battery; HCST = Health Canada Surveillance Tool; GWP = global warming potential; GHG= greenhouse gas; ***: *p* < 0.001; **: *p* ≤ 0.01; *: *p* ≤ 0.05; † = average computed from the absolute correlation values. μ ± σ: mean ± standard deviation of absolute values of correlation coefficients.

## Data Availability

Data on nutrients composition of foods has been provided by “Food Composition Database for Epidemiological Studies in Italy” (BDA-IEO) by Gnagnarella P, Salvini S, Parpinel M. Version 1.2015 available at the website http://www.bda-ieo.it/.

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
