# Peer review of "How Much Do Front-Of-Pack Labels Correlate with Food Environmental Impacts?"

_nutrients, 2023, doi:10.3390/nu15051176_

Round 1
Reviewer 1 Report
This study explores different food guides and front of pack food labels with respect to the environmental impact of the various recommendations.
I find the values given in Table 2 for the ITA GL recommendations to be suspicious. I would expect them to be similar to the Mediterranean diet, Nutriscore and the Canadian recommendations. In fact the values for the ITA GL have major differences with those 3 diets/recommendations but are quite similar to EAT Lancet. Even more surprising 2 sets of values (GWP and H20 footprint) are identical. These data do not seem consistent with Figure 5. This needs to be either corrected (if wrong) or discussed in the third paragraph of the Discussion.
I have gone over the paper and made many corrections in tracking.
I believe the paper contains valuable data. Unfortunately the presentation is so complicated that I can’t see the wood for the trees. I believe that most readers will be unable to understand the paper. The paper needs a major revision, especially the Results section. It is essential that the authors find a way to present the data so that the type of person who is likely to want to read the paper is able to understand it. This is absolutely essential.
Author Response
Dear Reviewer,
We greatly appreciated all your precise suggestions and the share of the document with tracking corrections that we all included in our revision.
METHODS SECTION:
- Lines 91-100: An introductory subsection (named 2.1) has been added to present the study design. “An introductory subsection (named 2.1) has been added to present the study design. “The study design rests on a stepwise approach. Firstly, a literature search has been conducted to collect food items from studies that assessed food Life Cycle Assessments (LCAs) for several environmental indicators. From the sum of all identified foods, a final food list has been extrapolated by removing duplicates and nonmatching foods with a food composition database. Then, eligible FOPLs and food scales have been identified and, according to their underpinning algorithms, ranks have been calculated for all listed foods to convert them in food scales. Lastly, a composite index has been developed to summarize environmental indicators and assess correlations with food scales.”
- Subsection 2.3 (Lines 101-113): The subsection 2.3 (Environmental impact of food products) was moved after 2.1 to explain what “Given the nature of the final foods lists in line 90 refers to” and give a clearer explanation of the methodology as follows: “The following step consisted of the creation of a database including all food items listed in the aforementioned papers, which resulted in a total of 500 food items selected.”
- Lines 115: a definition of LCA has been added.
- Lines 206-215: The subsection 2.7 was added to present the Mazziotta Pareto composite index. “A composite index has been developed to reduce the multidimensonal nature of sustainability by summarizing the eight environmental indicators into a single vector. The reduction of dimensions through the development of a single index allows to quantify, and therefore rank, the environmental impact of diets and FOPLs. The index was designed according to the Mazziotta Pareto Index protocol [25] and called Food Sustainability Mazziotta Pareto Index (FS-MPI). Data underwent a first normalization stage through a modified z-score. The normalized dataset was then aggregated by the sum of the arithmetic mean with a penalty function derived by the product of the standard deviation and the coefficient of variation.
- Subsection 2.8 (Lines 216-225): The statistical analysis subsection 2.8 was implemented to give more details about the analysis carried out.
RESULT SECTION: In order to make the data more understandable we modified the Results section as follows:
- We split subsection 3.1. Food scales correlation analysis into 3.1 Foods included in the analysis (lines 220-228) and 3.2 Food scale correlation analysis (lines 232-245)
- In subsection 3.1 we eliminated redundant results already presented in Tables 1.
- Subsection 3.2 was so reviewed (lines 240-253: “For each one of the eight scales of food healthiness (computed for the above 182 foods), the correlation with each one of the eight indicators of environmental sustainability have been computed according to the studies of Clune et al. (Global Warming Potential) [15], Poore et al. (Greenhouse Gas Emissions, Land Use, Acidification, Eutrophication, and Freshwater Withdrawal) [16], and Petersson et al. (Carbon Footprint and Water Footprint) [17]. The obtained correlation matrix has been reported in Table 2.
Healthier foods are associated to higher values of ITA-GL, Eat Lancet Diet, and HSR, and to lower values for MD, NS, IFLs, NIB, and HCST. It explains the reason of some inverse (negative) correlations with the environmental indicators, for which higher values correspond to higher environmental impacts. The last column and the last raw report the means (and standard deviations) of the absolute values of these correlations.
The eight food scales showed different correlations, ranging, as absolute values, from a maximum of 0.824 to a minimum of 0.030. Five scales showed an average correlation above 0.5. The majority of the correlations were found to be statistically significant.”
- In subsection 3.2 we removed Figured 2 and all the text describing those data that could confound the reader and, thus, do not add value to the Results section.
- Subsections 3.3. was mostly reviewed in the lines 268-278: “Figure 2 provides different visual inspections of the values of correlation matrix reported in Table 2. In Figure 2(a), a radar chart was carried out from the eight emission indicators correlated to the eight food scales mapped. External lines correspond to higher correlations and, as shown in the, three lines highlight strong correlations (MD, ITA GL, and EAT Lancet), two lines have moderate correlations (NIB and HCST), two lines with low correlations (NS and IFLs) and the most internal line has weak correlation (HSR).”,
- Subsections 3.4. was reviewed as follows (lines 295-298): “Between the two FOPL groups, FOPLs/portions showed higher correlations than FOPLs/100g for all the indicators. Finally, FOPLs/100g not only showed lower correlation values, but also higher variability (as highlighted by the wide standard deviations).”
- Subsection 3.5. was reviewed as follows (lines 300-312): The last step of the analysis was carried out using the Food Sustainability Mazziotta Pareto Index (FS-MPI), to summarize the eight environmental indicators into a single index. Even though the Mazziotta-Pareto Index is usually applied in the economics and human development fields of research [25-26], its solid background in the scientific literature makes it preferable to the new environmental index recently developed by Clark et al. [27]. However, if compared in a ranking order, scales have positioned themselves the same with both methods.
- Spearman correlations between FS-MPI and the eight food scales are shown in Figure 4. All of them are statistically significant, but Mediterranean Diet showed the highest score (R=0.738), then the EAT-Lancet and the Italian Dietary Guidelines had similar results (0.705 and 0.676, respectively), followed by NIB (0.497) and HCST (0.431), then by Israeli WLs and NS with very narrow differences (0.379 and 0.353, respectively), and lastly HSR (0.185) (Figure 4).
- We added statistical significance to the bar plot in Figure 4.
DISCUSSION SECTION:
- In regard to the ITA GL values in Table 2 and Figure 4, given the as coincidental as unexpected nature of the results we discussed them in the third paragraph of the Discussion section (lines 330-337): “Despite the differences in macronutrient percentages between the EAT-Lancet diet and the Italian Dietary Guidelines, the similarities observed in Table 2 may come from the fact that they both recommend low amounts of animal products and that their food scales are the only calculated by the product of food portions and food frequencies.”
- Strengths and limitations subsection was moved at the end of the discussion section.
Lastly, a language revision of the paper was carried out.
Kind regards

Reviewer 2 Report
The underlying aim of the paper is interesting, however the paper is somewhat difficult to read and the text is not easy to follow. I suggest a major revision to the paper whereby language is heavily edited and the design and methods (and the reasons behind their selection) are more clearly and specifically described.
Abstract: not clear what the paper is about (type of study, aim, etc).
Introduction: Some sentences are too long and complex (e.g. line 50-55). Also the aim is written in a complex manner. I would advise that the authors rewrite these long sentences in shorter and more simple and specific terms (particularly the aim).
Methods:
The design and methods (and the reasons behind the selection of these specific methods and design) are not clear. To help organize the section, I suggest that there are clear outlines for the design, methods/procedures and data. Currently only the analysis is clearly outlined, but also this section can go deeper stating this type of analysis was used to assess this type of relationship between X and Y variables.
Why did the authors only choose reviews for their study selections (and not go back to the original studies?) Why did the authors only choose one database to search for articles? Is this comprehensive enough? Also how do you know that the search string is comprehensive enough? Are 3 papers enough?
Not clear what final foods lists in line 90 refers to.
Line 108 missing verb. Sentence is incomprehensive
Please provide definition for new terms (e.g. Line 115 Life cycle assessment)
Results:
I suggest adding some interpretation to the results to facilitate reading.
It is difficult to comment on the results when the methods are not clear to me.
Discussion:
Line 321 ‘only one research has investigated their correlation with the impact of foods on the environment [27].’ But reference 27 is a qualitative narrative review study.
Author Response
Dear Reviewer,
We greatly appreciated all your precise suggestions, which we took in great consideration in our revision.
Abstract:
- Lines 14-18: we specified the aim of the research: “Since global climate change has recently become an urgent matter this paper aims to investigate the correlations between different food health scales, including some FOPLs currently adopted by one or more countries, and several sustainability indicators. For this purpose, a sustainable composite index has been developed to summarize the environmental indicators and compare food scales.”
Introduction:
- Lines 50-55: we modified the too complex sentences on lines 50-55 as follows: “A first classification divides them into directive and non-directive labels. Non-directive labels are informative labels that show the nutrient composition of foods and percentage amounts of nutrients within a standard daily dietary pattern. Differently, directive labels are the results of various algorithms used to warn and/or rank foods, with the goal of modifying individual eating behaviors by associating each food with a higher or lower level of health.”
- Lines 85-89: We reformulated the aim of the paper as follows: “The aim of this study is to understand whether and how much current FOPLs integrate healthiness and sustainability. In order to do so, this paper investigated the correlations between different food health scales, including some FOPLs currently adopted by one or more countries, and several sustainability indicators such as Carbon footprint, Land Use, and Water footprint.”
Methods:
- Our purpose was not to review systematically the literature but to create a database of food environmental impacts by gathering data from systematic reviews found in the literature.
- Only the reported 3 papers were found but as explained in the section 3.1 (lines 221-229), the final food list was enough in terms of food items and within-category food variability to give significant results.
- Lines 91-100: An introductory subsection (named 2.1) has been added to present the study design. “The study design rests on a stepwise approach. Firstly, a literature search has been conducted to collect food items from studies that assessed food Life Cycle Assessments (LCAs) for several environmental indicators. From the sum of all identified foods, a final food list has been extrapolated by removing duplicates and nonmatching foods with a food composition database. Then, eligible FOPLs and food scales have been identified and, according to their underpinning algorithms, ranks have been calculated for all listed foods to convert them in food scales. Lastly, a composite index has been developed to summarize environmental indicators and assess correlations with food scales.”
- Lines 101-113: The subsection 2.3 (Environmental impact of food products) was moved after 2.1 to explain what “Given the nature of the final foods lists in line 90 refers to” and give a clearer explanation of the methodology as follows: “The following step consisted of the creation of a database including all food items listed in the aforementioned papers, which resulted in a total of 500 food items selected.”
- Lines 115: a definition of LCA has been added.
- Lines 144-148: the missing verb was added, giving the right meaning to the sentence
- Lines 206-215: The subsection 2.7 was added to present the Mazziotta Pareto composite index. “A composite index has been developed to reduce the multidimensonal nature of sustainability by summarizing the 8 environmental indicators into a single vector. The reduction of dimensions through the development of a single index allows to quantify, and therefore rank, the environmental impact of diets and FOPLs. The index was designed according to the Mazziotta Pareto Index protocol [25] and called Food Sustainability Mazziotta Pareto Index (FS-MPI). Data underwent a first normalization stage through a modified z-score. The normalized dataset was then aggregated by the sum of the arithmetic mean with a penalty function derived by the product of the standard deviation and the coefficient of variation.
- Lines 216-225: The statistical analysis subsection 2.8 was implemented to give more details about the analysis carried out. “Data entry and analysis were carried out with the statistical software Jamovi (Ver. 2.3.18). A descriptive statistics analysis was carried out to study the behavior of all the variables, and then the Spearman correlation has been applied to find possible correlations between the 16 variables in the object. Furthermore, a hierarchical cluster analysis (parameter settings: distance measure = Euclidean; clustering method = warp D2) was performed on the correlation matrix to individuate patterns among food scales that could explain the observed relationships with environmental indicators. Finally, the FS-MPI was applied to the 8 environmental indicators to rank the overall environmental impact of diets and FOPLs using the composite index and assess which scale was more correlated”
Results:
- Lines 220-228: We split subsection 3.1. (Food scales correlation analysis) into 3.1 Foods included in the analysis and 3.2 Food scale correlation analysis
- We eliminated redundant results already presented in Tables 1 and 2.
- We removed Figured 2 and all the text describing those data that could confound the reader and, thus, do not add value to the Results section.
- We reviewed the subsections 3.3., 3.4., and 3.5. eliminating redundant results and reformulating some sentences to make them clearer to the reader (lines 260-270, lines 285-289, and lines 294-299).
- We added statistical significance to the bar plot in Figure 4.
Discussion:
- Reference [27] on Line 321 refers to the article of Clark et al., 2022 that is not a narrative review but estimates the environmental impacts of 57,000 food products.
Lastly, a language revision of the paper was carried out.
Kind regards

Round 2
Reviewer 1 Report
The paper has now been much improved. The content is now much clearer. However the quality of the writing still needs much work. This is essential in order to make the paper acceptable for publication. As before I have edited the paper using tracking. It is essential that the authors revise the paper based on my comments and suggestions.

Author Response
Dear Reviewer,
We greatly appreciated all your precise suggestions and the share of the document with tracking corrections that we all included in our revision. In particular, we modified the following paragraphs:
INTRODUCTION SECTION:
- Lines 49-50: Paragraph was reformulated inverting the order of FOPLS description “Directive labels use an algorithm that warns and/or ranks foods, with the goal of modifying individual eating behaviors by associating each food with different levels of healthfulness [5]. The most used models of directive labels are Ranking Labels, Traffic Light Labels, Warning Labels, and Endorsement Logos. By contrast, non-directive FOPLs (such as Reference Intakes and related labels), are informative labels that rely solely on nutrition facts. They show the amount of a small number of food components (typically fat, saturated fats, sugars, salt, and energy) in the food and relative (as percentage) amounts within a standard daily dietary pattern and. Some non-directive FOPLs also display the amounts that refer to a recommended serving size. Therefore, they do not associate any judgment with foods, and leave the consumer free to interpret the information provided. [6]”
- Lines 119-123: This sentence missed some words in your document. However, we reformulated as follows: “for instance, in the study by Poore et al. [16], units are expressed as Retail Weight Functional Units (FU) and include losses between distribution and retail but not consumer losses. Functional units are converted into Kg for solid commodities and into mL for liquid ones, as indicated in the present research.
DISCUSSION SECTION:
- Lines 347-349: The sentence was modified as follows: “For example, IFLs and NS were found to score similarly in average either in the correlation matrix (0.382±093 and 0.373±0.106, respectively) or in the FS-MPI analysis (0.379 and 0.353, respectively).”
Kind regards

Reviewer 2 Report
The design and results are clearer now, however the methods section (2.2 -2.7) can still use a little more specificity and better flow (e.g. perhaps also include what the purpose and role of each of the subsections is, or how the relate to each other). At the moment, this is still not easy to read.
Author Response
Dear Reviewer,
We greatly appreciated your suggestions, which we took in consideration in our revision as follows.
Methods: A brief introductory sentence was added to every subsection from 2.2 to 2.7 to facilitate the reading.
- Subsection 2.2. line 96: “The first stage of the research was to” was added; line103-107: “The following passage consisted of an overall grouping of all food items listed in the aforementioned manuscripts, which resulted in a total of 500 food items selected. Then, the list was examined to eliminate all beverages and non-edible commodities. Lastly, the detection and the elimination of duplicate items between the different papers were carried out, resulting in a final list of 182 food commodities.” was modified; line 111: “The final food list collected the impact values of” was added.
- Subsection 2.3. lines 125-126: “After the finalization of the food list, a search for nutrient profiling was carried out among those that are currently adopted by countries.” Has been added at the beginning of the subsection to connect it with the previous one and introduce the following. lines 142-143: “Then, food ranks were calculated for each FOPL as explained in subsection 2.5.” has been added at the end of the subsection to connect it to subsection 2.5.
- Subsection 2.4. lines 144-150: “In order to validate the results of the correlations between FOPLs and the environmental indicators, it was important to compare them to food scales that already include both healthiness and sustainability.” was added at the beginning of the subsection to explain the aim of this stage. And then “Thus, as a positive control of the research, validated healthy and sustainable diets, such as the Italian National Dietary Guidelines Recommended Diet (ITA-GL) [14], the EAT-Lancet Commission Diet (EAT Lancet) [11], and the Mediterranean Diet (MD) [12], have been selected and transformed into food scales (as described in section 2.6).” was implemented to connect it to Subsection 2.6.
- Subsection 2.5. lines 153-154: “Once the selection of FOPLS and heathy diets has been completed, the next stage was aimed at calculating the ranks of listed foods to obtain a food scale for each FOPL” was added to link it with the previous subsection and explain its purpose.
- Subsection 2.6. lines 191-193: “In contrast to FOPLs, healthy diets were transformed into food scales following the principle that the more the recommended amount of a food, the more it can be considered healthy.” was implemented and reformulated to improve the readability of the subsection.
- Subsection 2.7. lines 210-212: “The last stage consisted in the development of a composite index to reduce the multidimensonal nature of sustainability by summarizing the eight environmental indicators into a single vector.” Was implemented to conclude the description
Other minor changes were applied throughout the manuscript to improve its understandability.
Kind regards
